# Efficient Activation of Coal Fly Ash for Silica and Alumina Leaches and the Dependence of Pb(II) Removal Capacity on the Crystallization Conditions of Al-MCM-41

**DOI:** 10.3390/ijms22126540

**Published:** 2021-06-18

**Authors:** Xu Zhang, Tao Du, He Jia

**Affiliations:** 1State Environmental Protection Key Laboratory of Eco-Industry, Northeastern University, Shenyang 110819, China; 1310283@stu.neu.edu.cn (X.Z.); syxuzi@163.com (H.J.); 2Simulation Center, Shenyang Institute of Engineering, Shenyang 110136, China

**Keywords:** coal fly ash, CFA activation, Al-MCM-41, crystallization parameters, Pb(II) removal capacity

## Abstract

In this study, four different coal fly ashes (CFAs) were used as raw materials of silica and alumina for the preparation of the alumina-containing Mobil Composition of Matter No. 41 (Al-MCM-41) and the exploration of an activation strategy that is efficient and universal for various CFAs. Alkaline hydrothermal and alkaline fusion activations proceeded at different temperatures to determine the best treatment parameters. We controlled the pore structure and surface hydroxyl density of the CFA-derived Al-MCM-41 by changing the crystallization temperature and aging time. The products were characterized by small-angle X-ray diffraction, nitrogen isotherms, Fourier-transform infrared spectroscopy, ^29^Si silica magic-angle spinning nuclear magnetic resonance, and transmission electron microscopy, and they were then grafted with thiol groups to remove Pb(II) from aqueous solutions. This paper innovatively evaluates the CFA activation strategies using energy consumption analysis and determines the optimal activation methodology and parameters. This paper also unveils the effect of the crystallization condition of Al-MCM-41 on its subsequent Pb(II) removal capacity. The results show that the appropriate selection of crystallization parameters can considerably increase the removal capacity over Pb(II), providing a new path to tackle the ever-increasing concern of aquic heavy-metal pollution.

## 1. Introduction

Intensive anthropogenic activities, such as mining, metal processing, battery manufacturing, fertilizer, electroplating, textile, printing, and refining, are discharging a profusion of heavy metals into rivers, ponds, lakes, and reservoirs, posing a severe social concern. Lead (Pb(II)) is recognized as one of the most hazardous heavy metals due to its high toxicity, nonbiodegradability, and body accumulation. Long-term intake, even at trace amounts, irreversibly damages the central nervous system, kidney, brain, and reproductive system of human beings [1]. Accordingly, its concentration for drinking water is strictly restricted below 0.1 mg/L by the World Health Organization and even below 0.05 mg/L by some developed countries [2]. Efficient Pb(II) removal technologies are indispensable to fulfill the above standard. They include ion exchange, chemical precipitation, membrane filtration, solvent extraction, and reverse osmosis [3]. Adsorption prospers among numerous methods due to its low cost, simplicity, flexibility, and easy manipulation.

The selection of adsorbent is vital for adsorption. Generally, a promising adsorbent should have a low cost, high capacity, fast kinetics, high selectivity, mild regeneration condition, and high recyclability. Activated carbon is the most commonly used adsorbent for wastewater decontamination and indoor air purification [4]. However, its widely distributed pore size lowers its target selectivity and removal efficiency. In addition, the ever-increasing price impels researchers to search for alternatives. Zeolites, being aluminosilicate crystals, have also been extensively investigated for heavy-metal removal due to the high cation exchange capacity, the low cost and bulk availability of raw materials, and excellent thermal, hydrothermal, and mechanical stability [5]. However, the microscale pore size (<2 nm) limits its adsorption over macromolecules. Ordered mesoporous silicas have gained an increasing amount of consideration in this sphere. In 1992, scientists from Mobil used self-assembled supramolecular surfactants to synthesize a series of ordered mesoporous silicas designated as M41S (MCM-41 series) [6,7]. The numbers of the M41S family include the two-dimensional (2D) hexagonal MCM-41 (Mobil Composition of Matter No. 41), the cubic MCM-48 (Mobil Composition of Matter No. 48), and the lamellar MCM-50 (Mobil Composition of Matter No. 50). The emergence of the M41S family has triggered a boom in discovering more ordered silicas, such as the 2D rectangular SBA-8 (Santa Barbara Amorphous No. 8), the large-pore 2D hexagonal SBA-15 (Santa Barbara Amorphous No. 15), and HMS (Hexagonal Mesoporous Silica) [8]. MCM-41 is one of the most studied among them, discovered by the Mobil Corporation in 1992 [7]. Its physical advantages, for instance, a well-defined hexagonal geometry, an open pore structure, a narrowly distributed and adjustable nanoscale pore size, a high specific surface area, a large pore volume, a substitutable amorphous silica skeleton, and an easily modified surface, make it an outstanding adsorbent meeting almost all conditions [8].

Typically, MCM-41 itself shows negligible heavy-metal removal capacity due to its nonpolar surface [9]. However, its integration with functional ligands, such as amine, thiol, amide, phosphate, and carboxylates [10], possesses a high capacity and target selectivity. For instance, Li et al. [11] used thiol-functionalized MCM-41 to capture Sb(III) from aqueous media to achieve an uptake of 164.8 mg/g. Deng et al. [12] fabricated hollow SiO_2_ microspheres with surfaces rich in thiol groups for Pb(II) removal and finished with a maximal adsorption capacity of ca. 125.3 mg/g. Benhamou et al. [13] employed amine-functionalized pore-expanded MCM-41 and MCM-48 for the complexation of Cd(II), Co(II), Cu(II), and Pb(II), giving the affinity order Cu(II) > Pb(II) > Cd(II) > Co(II). Post-synthesis grafting is one main approaches for immobilizing functionalities to the silica wall. The binding between host and ligand is mighty, and the complex is hydrothermally stable and structurally ordered [14]. The hydroxyl groups present on the host wall serve for the grafting between hosts and ligands. Thus, the hydroxyl surface density determines the number of functional groups attached and the level of heavy-metal removal capacity. Manu et al. [15] controlled the concentration of hydroxyl groups of silica gel by varying calcination temperatures. Results indicated that the number of hydroxyl groups decreased with increasing temperature, which led to a decrease in the number of aminopropyl groups grafted and the subsequent Cu(II) removal capacity.

Amorphous silicates in a solution can crystallize into MCM-41 under a wide range of conditions. During the synthesis, after the MCM-41 stock solution is prepared, it is usually transferred to a Teflon-lined stainless-steel autoclave for crystallization at certain temperatures (20–150 °C) and times (12–72 h). It can boost the silicate condensation around the preliminarily formed hexagonal array, giving rise to more structurally stable and ordered MCM-41. The crystallization temperature and aging time are the two predominant parameters of MCM-41 crystallization. Increasing crystallization temperature or prolonging aging time can increase MCM-41 crystallinity and abate structural deficiencies [16,17]. However, the attachable hydroxyl groups will decrease in number, as evidenced by Cheng et al. [18] and Manu et al. [15]. As a result, the heavy metal (such as lead, copper, cadmium, mercury, nickel, and cobalt) removal of the derived adsorbent will be inevitably affected. For instance, McManamon et al. [19] derived different removal capacities over lead and cadmium by changing the crystallization temperature of SBA-15.

The crystallization process produces a significant effect on the structural characteristics of the produced MCM-41. The pore size and its distribution are the structural parameters most closely related to adsorption [20]. Large and ordered pores allow for the grafting of massive ligands while keeping good molecule diffusivity. Specific surface area and pore volume are also closely related to adsorption. High surface area and large pore volume are the structural parameters we usually want to obtain. As crystallization parameters change, these structural parameters closely related to adsorption also change, influencing the adsorption capacity of the derived adsorbents. For instance, with increasing crystallization temperature, the synthesized MCM-41 increases in structural uniformity but decreases in specific area [16]. Therefore, the crystallization process following the prepared MCM-41 stock solution is vital to the subsequent removal capacity of heavy metals. However, relevant reports are scanty.

Another area of extensive concern in synthesizing mesoporous silicas is to employ low-cost silica-containing sources. Up to now, researchers have successfully synthesized MCM-41 using a diversity of silica-containing precursors, such as (i) natural minerals (natural clay [21], diatomite [22], kaolin [23], and so on), and (ii) anthropogenic solid wastes (rice husk ash [24], peanut shell ash [25], wheat straw ash [26], coal slag [27], coal fly ash [28], and so on). Coal fly ash (CFA) is a byproduct of coal-fired power plants, originating from the nonflammable ashes in coal, and it is collected by precipitators from flue gas. As the world’s largest coal consumer, China’s annual CFA product exceeds 500 million tons. Faced with such tremendous CFA production, although the utilization rate of CFA in China is as high as 80%, which concentrates in traditional fields such as building materials, road construction, and commercial concrete [29], there remains about 100 million tons of idle CFA every year. By the end of 2020, the total idle CFA in China reached an astonishing two billion tons, becoming one of the largest amounts of solid waste in China, occupying many land resources and polluting the local environment. Under the social background of increasing environmental protection pressure, the disposal of CFA solid waste is becoming more urgent and significant. However, CFA is a worthy alternative to conventional adsorbents in water decontamination. Zanoletti et al. [30] employed CFA to remove anionic surfactants from water, achieving a removal efficiency as high as 96%. More importantly, the embodied energy is almost two orders of magnification lower than the frequently used adsorbent, activated carbon. This illustrates that the application of CFA to treat contaminated water is environmentally sustainable. In addition to being directly used as an adsorbent for water purification, CFA can act as raw material to prepare ordered aluminosilicates and subsequently treat wastewater. The silica (SiO_2_) content of CFA is approximately 60–70 wt.%, while that of alumina (Al_2_O_3_) is ca. 20–30 wt.% [31]. Therefore, this enables making ordered aluminosilicates out of CFA. The popularization of the application in this area will relieve the increasing disposal pressure of CFA solid waste to a certain extent.

The first step in using CFA to prepare ordered aluminosilicates is to extract the silica and alumina in CFA, and this process is called the activation of CFA. Alkaline hydrothermal and alkaline fusion treatments are the two most used CFA activation strategies [5]. Alkaline hydrothermal activation of CFA was used originally for preparing zeolite by Holler in 1985 [32]. The activation process can be either atmospheric or pressurized, and both have been verified as effective. The activated solution directly crystallizes into zeolite under proper synthesis conditions without further treatment. However, before synthesizing ordered mesoporous silicas, the subsequent separation of the solution after activation is required to obtain the silica- and alumina-containing supernatant. Alkaline fusion activation of CFA was first used for the preparation of Na-X zeolite by Shigemoto et al. [33]. It yielded the stable crystalline structure of mullite and improved the availability of CFA. The crystallinity of Na-X zeolite peaked under the following conditions: (i) calcination temperature of 550 °C, (ii) calcination time of 1 h, and (iii) NaOH/CFA mass ratio of 1.2. Hence, almost all researchers who successfully used CFA to synthesize MCM-41 have used this condition [34]. It is noteworthy that 550 °C is the optimal condition to form Na-X zeolite rather than MCM-41. It seems more significant for MCM-41 preparation to derive more extracts of amorphous silica. As elucidated by the authors [33], the acid-soluble components increased with rising fusion temperature. Hence, a higher fusion temperature may be more effective in preparing MCM-41. However, studies have focused on the possibility of using CFA to prepare MCM-41 but not on how to make the best use of CFA. In addition, CFA is extremely diverse in chemical composition and crystalline phases [28]. The conclusion from an individual CFA may not have generality, but most relevant studies have disregarded this point.

The purpose of this work was to determine an efficient, cost-effective, and universal CFA activation strategy by varying CFA types, activation methods, and process parameters. The CFA-derived supernatants were crystallized into Al-MCM-41 at different temperatures (30–150 °C) and times (24–72 h). Through batch experiments, we evaluated the influence of the crystallization condition of MCM-41 on the subsequent removal capacity of Pb(II), unveiling a new path to promote heavy-metal removal.

## 2. Results and Discussion

### 2.1. CFA Activation

#### 2.1.1. Raw CFA Analysis

Table 1 shows the main chemical composition of CFA1–4 employed in this study. In all CFAs, SiO_2_ was the most predominant component with a proportion of 58.9–64.3 wt.%, followed by Al_2_O_3_, accounting for 19.8–22.7 wt.% and then Fe_2_O_3_, CaO, K_2_O, and MgO. It is clear that, although CFA1–4 were from different coal sources and had various formation conditions, there was no significant difference in their chemical compositions. Meanwhile, the rich content of silica and alumina enabled preparing Al-MCM-41 from CFA.

Figure 1 presents the XRD patterns of CFA1–4. The major crystalline phases for CFA1–3, i.e., the PC ashes, were quartz and mullite, together with some minor phases such as hematite and berlinite for 1 and 3, and only hematite for 2. However, quartz became the solely predominant crystalline phase for CFA4, i.e., the CFB ash, and only one diffraction peak matched the mullite PDF pattern. The generation of mullite requires a comparatively high temperature (>1000 °C), but the CFB furnace temperature cannot reach that level [35]. Hence, there was a noticeable difference in crystal phase composition between PC and CFB ashes. In addition, we could observe different degrees of bumps, i.e., wide-angle peaks, at around 2*θ* = 25°, illustrating the presence of amorphous aluminosilicate glasses [28].

The quantitative analysis of XRD gave the mass ratio of quartz and mullite. The content of quartz in CFA1–4 differed little (from 38.3–44.7 wt.%). However, the mass ratio of mullite in CFA4 was only 9.1 wt.%, much lower than in CFA1–3 (19.6–25.3 wt.%). The mass ratio of amorphous SiO_2_ and Al_2_O_3_ glasses, determined by excluding the crystalline phases from the bulk compositions, is shown in Table 1.

#### 2.1.2. Activation Optimization of CFA

Table 2 lists the activation methodologies and process parameters over CFA and the results obtained. Regarding the hydrothermal activation of CFA, as the temperature increased from 100 to 200 °C, the conversion rate of silica decreased for CFA1, but it increased for CFA2–4, and all alumina conversion rates decreased. Therefore, we could not draw a universal conclusion on how alkaline hydrothermal temperature influences the extraction of silica.

However, not all amorphous SiO_2_ and Al_2_O_3_ glasses were transferred to silica and alumina ions in supernatants; the conversion rate was 33–67% for silica and 13–23% for alumina. There were still unconverted aluminosilicate glasses, which is consistent with previous reports [36,37]. Furthermore, a long time of hydrothermal treatment may be disadvantageous for the transfer of Al_2_O_3_ glasses to alumina ions since the irreversible conversion process may become more active as time increases [38]. Therefore, the conversion rate of alumina ions is universally lower than that of silica.

Figure 2 presents the XRD patterns of CFA1–4 residues after activation. As shown in Figure 2a,c,e,g, the original XRD patterns of CFA1–4 did not change significantly after 100 °C of hydrothermal treatment, except for the wide-angle peaks at around 2*θ* = 25° decreasing dramatically. This indicates that atmospheric hydrothermal treatment can extract amorphous aluminosilicates from CFAs but cannot open the structure of quartz and mullite in the span of 3 h. Likewise, before high-temperature fusion activation of CFA, Shoppert et al. [39] leached the amorphous aluminosilicates in CFA using 90 °C atmospheric alkaline hydrothermal treatment. When the temperature increased to 200 °C (corresponding pressure of ca. 1.55 MPa), many original diffraction peaks disappeared along with the appearance of several additional ones ascribed to zeolites. They were analcime, sodalite, cancrinite, zeolite P, and zeolite I. This illustrates that the pressured hydrothermal treatment can extract silica and alumina from crystals. Simultaneously, the leaches partially crystallize into zeolites without being transferred to the supernatant. Querol et al. [40] reported that relatively high temperatures (125–200 °C) accelerated the leaching of silica and alumina from CFA particles, and the leaches quickly crystallized to zeolites, which were the same as those formed in this study. Therefore, the extraction effect of pressurized alkaline hydrothermal activation of CFA is limited by the rapid formation of zeolite under those conditions. This explains why a higher temperature in favor of leaching more silica and alumina substances from CFA, in some cases, cannot yield a supernatant with a higher concentration of silica and alumina.

To further enhance the effect of hydrothermal activation, we introduced microwave to the hydrothermal treatment, as shown in Figure 2i. Studies [41,42] have indicated that microwave-assisted hydrothermal treatment sped up the leaching of silica and alumina from CFA and significantly shortened the time of zeolite synthesis. We used the atmospheric microwave-assisted extraction method to activate CFA at 100 °C for 1 and 3 h, respectively. When the activation time was 1 h, the conversion of silica and alumina was higher than that without microwave assistance for 3 h, as shown in Table 2. The XRD patterns differed little, as shown in Figure 2i, but the intensity of peaks after microwave-assisted activation was universally lower. Therefore, microwaves made the activation of CFA more efficient. This is advantageous for the rapid and even heating of a solution since water molecules absorb microwave irradiation directly, not via heat transfer [38]. When heating for 3 h, the intensity of peaks further decreased. We also observed peaks ascribed to Na_2_SiO_3_, likely due to insufficient stirring. However, the conversion of silica and alumina still increased, yielding a conversion rate of more than 18.9% for the former, related to the content of amorphous SiO_2_ glasses. This confirmed the leaching of silica from crystals.

As for the high-temperature alkaline fusion activation of CFA, the silica and alumina leaches were positively correlated with temperature for CFA1–4. As shown in Table 2, as the fusion temperature increased from 550 to 650 °C, the relative increments in silica and alumina conversion rate were 22.4% and 65.1%, respectively. With a further 100 °C increase in temperature, they increased again by 6.8% and 46%, respectively. Since silica is universally much more leached than alumina, the total leach increment from 550 to 650 °C was far superior to that from 650 to 750 °C. To comprehensively evaluate the influence of alkaline fusion temperature on the extraction effect, we also considered the energy consumption. We used an electric heating furnace as an example to quantitatively assess the relationship between the extraction effect and the energy consumption and determined the optimal activation conditions, as shown in Table 3.

It is clear from Table 3 that, for both CFA1 and 4, the 650 °C alkaline fusion treatment yielded the lowest energy consumption rate *q*; this was also true for CFA 2 and 3, but that information is not listed here due to space limitations. Therefore, combined with energy consumption analysis, we obtained the universally optimal alkaline fusion temperature, 650 °C, at which we could obtain the maximal yield of Al-MCM-41 when consuming the same amount of energy.

Compared to hydrothermal activation without microwave assistance, conversions of both silica and alumina were increased severalfold. As shown in Figure 2b,d,f,h, most of the original diffraction peaks disappeared, illustrating that high-temperature alkaline fusion can derive silica and alumina not only from the amorphous aluminosilicates but also from the crystalline phases. Meanwhile, several additional diffraction peaks were observed belonging to sodalite and cordierite. Shoppert et al. [39] also obtained sodalite from the XRD patterns of calcined CFA residues when fusion temperature was higher than 200 °C. However, in this work, the intensities of these peaks after alkaline fusion activation were weaker than those after alkaline hydrothermal treatment, especially when the fusion temperature was 650 and 750 °C. This illustrates that alkaline fusion activation can transfer more leaches to the supernatant. Izabela et al. [35] made a similar comparison and concluded that 500 °C alkaline fusion yielded more extracts than 100 °C alkaline hydrothermal treatment.

Although we determined cost-effective and efficient CFA activation conditions, the maximal silica conversion rate was only 28.8%. To explain this, we evaluated the XRD pattern of CFA2 residue after 650 °C alkaline fusion treatment. Figure 1e shows the peaks of sodalite (3NaAlSiO_4_·NaCl) and cordierite (Al_3_(Mg, Fe)_2_[Si_5_AlO_18_]), which contain a large mass of silica and alumina. Therefore, many extracts of silica and alumina from CFA were transferred to these high silica/alumina-containing materials, leading to a low conversion rate. Another important reason that led to a low conversion rate is related to the particle size of CFA. As shown in Figure 3, the particles had a micron size. It is highly possible that the reaction of alkali and CFA happens mainly at the surface, not deep inside, resulting in the incomplete participation of CFA.

We also estimated the conversion rate of previous studies for comparison, as shown in Table 4. The conversion rates of silica and alumina from CFA in previous works were also low. However, we determined a comparatively better activation parameter (650 °C), which decreased the energy consumption rate by ca. 10% compared to the frequently used one (550 °C).

Figure 3 presents the SEM images of CFA1–4 and the residues after 650 °C of fusion. Spherical particles can be observed for CFA1–3, i.e., PC boiler ashes. The particle size ranged from ca. 1 to more than 10 μm, in line with [45,46]. After alkaline fusion treatment at 650 °C, the particles became irregular bulks, and the particle surface was not smooth. This suggests that the reaction of alkali and CFA occurred on the surface [28]. However, there were only a few spherical particles for CFA4, i.e., CFB boiler ash. Most particles showed irregular shapes since the relatively low furnace temperature of the CFB boiler could not melt the ashes in coals [29].

### 2.2. Relevance of Crystallization Parameters and the Al-MCM-41 Structure

The names of samples crystallized at different temperatures and times were listed in Section 3.4. Figure 4 compares the SXRD patterns of Al-MCM-41 obtained by varying (a) the crystallization temperature, (b) the aging time, (c) the template removal strategies (calcination or solvent extraction), and (d) the surface nature (with or without grafting). As shown in Figure 4a, when the aging time was maintained constant at 48 h, there were at least three *hk0* diffraction peaks for all samples. This indicates that all products that crystallized at different temperatures had MCM-41 type structures. As crystallization temperature increased, the peak intensities grew, and (100) slightly shifted toward a lower 2*θ*. These phenomena illustrate that high crystallization temperatures could produce Al-MCM-41 with a higher degree of order and a larger basal space *d*_100_ [16]. Likewise, the effect of aging time on the structure of Al-MCM-41 was vital. Figure 4b shows that, when keeping the crystallization temperature constant at 120 °C, prolonging the aging time could promote the order degree of Al-MCM-41 and increase *d*_100_. However, as the aging duration increased, the variation of SXRD patterns from 48 to 72 h was not as noticeable as from 24 to 48 h. This illustrates that, when keeping the crystallization temperature at ca. 100 °C, a longer hydrothermal treatment after 48 h produced a minor effect on the structure, which concurs with the study of Cheng et al. [18].

When using calcination to remove the CTA^+^ templates, as shown in Figure 4c, the calcined product (AM41C-120-48) had a more ordered structure but a narrower *d*_100_ compared to the extracted one (AM41S-120-48). This is because calcination could remove the template more efficiently than solvent extraction but led to lattice contraction [47].

Upon ligand (MPTMS and APTMS) grafting on AM41E-120-48, as shown in Figure 4d, three *hk0* diffraction peaks were still visible but decreased in intensities, illustrating the successful ligand grafting and still intact mesoporous structure [48].

As shown in Figure 5a,b, all samples prepared at different crystallization temperatures and aging times showed type IV isotherms, as characterized by three distinct adsorption stages: (i) a fast monolayer or multilayer adsorption at *p*/*p*_0_ < 0.2, (ii) a capillary condensation stage at *p*/*p*_0_ = 0.2–0.4, and (iii) a gradually increasing adsorption associated with external filling at *p*/*p*_0_ > 0.4 [35]. These features further validated the presence of an ordered mesoporous framework. As evidenced by the embedded illustrations (a) and (b), the pore sizes were mainly concentrated in the range of 3–4.5 nm. However, AM41ES-30-48 had a comparatively wide pore size distribution, which matched the characteristics of the isotherms with a gentle slope of capillary condensation and an unapparent inflection point.

Compared to solvent extraction, calcination could remove the template more thoroughly. Therefore, the calcined product (AM41C-120-48) had a larger pore volume, as shown in Figure 5c. As for ligand grafting, as shown in Figure 5d, the traits of type IV isotherms were still legible. However, the pore volumes dramatically decreased due to the pore filling.

Table 5 lists the pore structural properties obtained by SXRD and N_2_ isotherms. It is clear that, with increasing crystallization temperature (30–120 °C) and aging time (24–72 h), the specific surface area *S_BET_* decreased while the wall thickness W increased. A higher crystallization temperature and longer aging time facilitated the polymerization of silicate species around surfactant micelles, leading to a thicker W but a reduced *S_BET_* [16,18].

Figure 6 presents the TEM image of AM41E-120-48. There were honeycomb-like hexagonal pores in Region A1 and parallel lines in Region A2 when the electron beam illuminated the pores vertically and parallelly, respectively. This confirmed the generation of the MCM-41 structure [17,26]. We selected five arrays of pores in Region A1 and six in Region A2 to evaluate the basal spaces using the Digital Micrograph tool. They were, respectively, 4.13 and 4.27 nm, similar to the SXRD results (4.14 nm).

Figure 7 lists the SEM images of Al-MCM-41 synthesized at different crystallization temperatures and times. We can see honeycomb-like mesopores in each crystallization condition. However, many pores surrounded by the red rectangles in Figure 7g,h were distorted and showed worm-like shapes, illustrating that a noticeable decrease in pore order occurred when the crystallization temperature was 30 °C.

Figure 8 depicts the FTIR patterns for AM41E-120-48, AM41ES-120-48, and AM41A-120-48. They all presented characteristic bands ascribed to the silica–oxygen skeleton, namely, peaks at ca. 1068, 797, and 467 cm^−1^, caused by the asymmetric and symmetric stretching vibrations and bending vibration of siloxane Si–O–Si, respectively. The vibrational bands at ca. 1630 and 3500 cm^−1^ were due to the adsorbed water and surface silanol Si–OH [49]. After ligand grafting, we observed vibration bands at 2561 and 1550 cm^−1^, which belonged to thiol and amino groups, respectively [50,51]. In addition, the vibration band at ca. 2930 cm^−1^ (or 2940 cm^−1^) was attributed to C–H asymmetric stretching vibrations, and that at ca. 1468 cm^−1^ (or 1490 cm^−1^) was ascribed to the CH_2_ deformation vibration. These hallmarks confirm the successful grafting of MPTMS and APTMS.

Figure 9a,b show the ^29^Si NMR spectra of Al-MCM-41 crystallized at different temperatures and times. Each sample had two noticeable resonances at −102 (or −101) and −112 (or −111) ppm, which were attributed to the siloxane binding surroundings without hydroxyl groups [Si(OSi)_4_ (Q^4^)] and with isolated silanes [Si(OSi)_3_(OH) (Q^3^)], respectively. We obtained the areas under Q^4^ and Q^3^ by deconvolution, as shown in Table 6. With increasing crystallization temperature and aging time, the value of Q^4^/Q^3^ increased. This illustrates that a higher temperature and longer time promoted the polymerization of silanol, forming more Q^4^ units at the cost of Q^3^, which concurs with findings from Cheng et al. [18]. When using calcination to remove the template, as shown in Figure 9c, the value of Q^4^/Q^3^ increased sharply due to high-temperature dehydration. Upon ligand grafting, we also observed two additional resonances at ca. −60 and −70 ppm, as shown in Figure 9d. They were attributed to SiR(OSi)_3_ (T^3^) and SiR(OSi)_2_(OH) (T^2^), indicating the successful bonding of organic ligands to the silica host [52].

### 2.3. Relevance of Crystallization Parameters and Pb(II) Removal Capacity

Solution pH is vital in heavy-metal removal since it significantly affects the surface charge of adsorbents and its subsequent binding with heavy metals. Figure 10 lists the adsorption capacity of AM41ES-120-48 and AM41EA-120-48 toward Pb(II) in the pH range of 2–7. At a pH lower than 3, both aminated and thiolated Al-MCM-41 presented extremely low Pb(II) adsorption capacities. In this case, most affinity sites (–OH, –SH, or –NH_2_) were protonated and, thus, lost the ability to complex with heavy metals. As pH increased to 5, both uptakes increased distinctly due to less competition with H^+^ and an impaired repulsive force between the positively charges adsorbent surface and Pb(II). With a further increase in pH to 9, both uptakes still increased. However, the increments were not as visible as those in the pH range from 3 to 5. When pH increased beyond 6, many more hydroxide precipitations were formed, as elucidated by Stathi et al. [53]. This contributed to an increase in Pb(II) removal capacity but affected the adsorption. We determined that the optimal pH was 6 for both, as there was neither protonation of affinity sites nor hydroxide precipitation in this case, which concurs with [12,53,54]. Furthermore, thiol-functionalized Al-MCM-41 had superior Pb(II) removal capacity to its amine counterpart, consistent with the hard and soft acids and bases (HSAB) theory [55].

Figure 11a,b show the Pb(II) removal capacities of adsorbents built on the Al-MCM-41 crystallized at different temperatures and times. We can draw two conclusions. First, when keeping the crystallization temperature constant at 120 °C, a longer aging time was detrimental to the subsequent adsorption. Second, as crystallization temperature increased from 30 to 120 °C and aging time was maintained invariably at 48 h, the adsorption capacity presented an initial increase followed by a decrease, which peaked at 60 °C. To explain the above results, we quantitatively estimated the thiol proportion in adsorbents using XRF, as shown in Table 7. The removal capacity of Pb(II) was relevant to the number of thiol groups grafted. AM41ES-60-48 had the maximal thiol sharing of 10.32 wt.%, and it achieved the maximal adsorption capacity of 153.0 mg/g.

The maximal thiol loading rate of AM41ES-60-48 largely depended on the high surface hydroxyl density of AM41E-60-48 [15]. As shown in Table 6, AM41E-60-48 has the second-lowest Q^4^/Q^3^ with a value of 1.2; hence, it had a comparatively high surface hydroxyl density compared to all other Al-MCM-41, which provided a profusion of points used for grafting. However, the surface hydroxyl density was not the sole factor determining the number of functional groups grafted. AM41E-30-48 had the lowest Q^4^/Q^3^ of 0.9; thus, it should have had the highest number of hydroxyl groups, but it achieved a grafting number much lower than that of AM41E-60-48. This can be explained by the results of SXRD and N_2_ isotherms. More specifically, as crystallization temperature decreased to 30 °C, the synthesized Al-MCM-41 (AM41E-30-48) dramatically deteriorated in pore order. In this case, a blockage near the pore mouse during grafting probably occurs, limiting the accessibility of internal hydroxyl groups and leading to a decrease in the grafting number [20]. Walcarius et al. [54] reported that ordered mesoporous structures with a pore size of 3.5 nm showed an internal accessibility similar to that of large-pore silica gel with an average pore size of 6–7 nm. Therefore, an ordered pore structure is indispensable for MCM-41-type adsorbents to achieve a high metal removal capacity.

Due to the superior Pb(II) removal capacity of AM41ES-120-24 compared to AM41ES-120-48, AM41ES-60-24 was expected to present a higher uptake than AM41ES-60-48. Therefore, we also used AM41ES-60-24 to remove Pb(II) under the following conditions: adsorption temperature *T* of 25 °C, adsorbent dose *m* of 50 mg, solution volume *V* of 50 mL, contact time *t* of 200 min, pH 6, and initial Pb(II) concentration of 200 mg/L. However, AM41ES-60-24 did not exhibit the expected capacity at only 96.3 mg/g, much lower than AM41ES-60-48. The N_2_ isotherms indicated that AM41ES-60-24 had a poor pore order compared to its counterpart, as shown in Figure 12. This again validated the significance of pore order in the subsequent adsorption. It also illustrated that low-temperature crystallization requires a longer aging time to create an ordered pore structure than it does at a high temperature.

Figure 11c illustrates the effect of the template removal strategy on the subsequent Pb(II) removal capacity. AM41CS-120-48 had a far lower uptake than AM41ES-120-48. High-temperature calcination remarkably decreased the surface hydroxyl density compared to solvent extraction, with a Q^4^/Q^3^ of 4.1 versus 1.8, respectively. Therefore, it was detrimental to subsequent grafting, which is in line with the conclusion of Lee et al. [56].

For the thiol-functionalized Al-MCM-41, the number of affinity sites required for each Pb(II) ion ranged from 4.15–4.42, which agrees well with [50,55]. As the fraction of thiol groups grafted increased, the molar ratio of S/Pb(II) slightly decreased. Compared to the amino-functionalized Al-MCM-41 (AM41EA-120-48), thiol groups were the more efficient affinity sites to Pb(II) (S/Pb = 4.26 versus N/Pb = 9.49), thus presenting a universal high uptake over Pb(II), as shown in Figure 11d.

Langmuir and Freundlich isothermal models were also employed to characterize the adsorption process, as shown in Equations (1) and (2), respectively.
(1)$$q_{e}=q_{m}\frac{bC_{e}}{1+bC_{e}},$$
(2)$$q_{e}=k_{F}C_{e}{}^{1/n},$$
where *C_e_* (mg/L) is the equilibrium concentration of Pb(II), *q_e_* (mg/g) is the equilibrium adsorption capacity over Pb(II), *q_m_* (mg/g) is the maximal adsorption capacity over Pb(II), *b* is the Langmuir constant, and *k_F_* (mg/g(L/mg)^1/n^) and *n* are the Freundlich constants.

We can know the reaction favorability via Freundlich constant *n* or separation factor *R_L_*; the expression of *R_L_* is shown in Equation (3).
(3)$$R_{L}=\frac{1}{1+bC_{0}},$$

Generally, a value of 0.1 < 1/*n* < 0.5 indicates favorable adsorption, while a value of 1/*n* > 2 indicates unfavorable adsorption. *R_L_* can indicate unfavorable (*R_L_* > 1), linear (*R_L_* = 1), favorable (*R_L_* < 1), or irreversible adsorption (*R_L_* = 0) [22].

Table 8 lists the parameters obtained from Langmuir and Freundlich models. According to the observed deviation between experiment data and fitted curves and the value of R^2^, the Langmuir model yielded a better description of all adsorbents, except for AM41ES-30-48. The highly precise fit by the Langmuir model illustrated the monolayer adsorption of Pb(II) and the homogeneous distribution of affinity sites [57]. However, for AM41ES-30-48, the Freundlich model, indicating the heterogeneously dispersed affinity sites, was more suitable. This was probably because the poor pore order of AM41E-30-48, as characterized by SXRD and N_2_ isotherms, led to the uneven distribution of the affinity sites of AM41ES-30-48. Hence, either the Langmuir or the Freundlich model could present a better fitting, mainly depending on the distribution of affinity sites. Both cases have been shown [57,58]. In addition, all values of *R_L_* < 1 and 0.1 < 1/*n* < 0.5 illustrated that the adsorption of Pb(II) on silica composites was prone to occur.

A thermodynamic study was conducted to uncover the adsorption mechanism using the best adsorbent AM41ES-60-48, as shown in Figure 13. We employed the Van ’t Hoff equation to obtain the thermodynamic parameters, as expressed below [22].
(4)$$K_{d}=\frac{q_{e}}{C_{e}},$$
(5)$$\Delta G^{0}=-RT\ln K_{d},$$
(6)$$\ln K_{d}=\frac{\Delta S^{0}}{R}-\frac{\Delta H^{0}}{RT},$$
where *R* (8.314 J/(mol·K)) is the universal gas constant, *K_d_* (L/mg) is the distribution coefficient, Δ*G*^0^ (kJ/mol) is the Gibbs free energy change, Δ*H*^0^ (kJ/mol) is the enthalpy change, Δ*S*^0^ (kJ/(mol·K)) is the entropy change, and *T* (K) is the Kelvin temperature.

All values of Δ*G*^0^ were negative, indicating spontaneous adsorption. The positive value of Δ*H*^0^ (20 kJ/mol) revealed the endothermal nature of the reaction. Although the coordination of Pb(II) was exothermal, the preceding diffusion was endothermal. The endothermic value exceeded that of exothermicity, thus presenting an endothermal characteristic overall. This concurs with [10,59,60]. The positive value of Δ*S*^0^ (101 kJ/mol) illustrated increased randomness upon adsorption, despite losing a degree of freedom.

We chose several representative adsorbents to evaluate the time dependence of the Pb(II) removal capacity. The samples contained the best-performing adsorbent and those obtained using different ligands and template removal strategies: AM41ES-120-48, AM41CS-120-48, AM41EA-120-48, and AM41ES-60-48. All adsorbents presented two-stage adsorption behavior, i.e., fast initial adsorption and slow equilibrium adsorption, as shown in Figure 14a, which conforms to adsorption inherence [22]. The rapid initial adsorption of the first stage was due to a profusion of vacant affinity sites and a high solute concentration gradient. The uptake soared in this stage, accomplishing most of the total capacity. As unoccupied affinity sites dwindled, the adsorption rate decreased, entering the slow equilibrium stage. There were hardly any changes in uptakes after 100 min, revealing that the adsorptions reached equilibrium. Hence, a span of 200 min was sufficient for adsorption in this study.

As for the kinetics evaluation, we employed the pseudo-first-order, pseudo-second-order, and intraparticle diffusion models. The mathematical expressions are as follows:(7)$$q_{t}=q_{e}[1-e^{-k_{1}t}],$$
(8)$$q_{t}=\frac{k_{2}q_{e}{}^{2}t}{1+k_{2}q_{e}t},$$
(9)$$q_{t}=k_{i}t^{0.5}+c,$$
where *t* (min) is the contact time, *q_t_* (mg/g) and *q_e_* (mg/g) are the uptakes of Pb(II) at time *t* and equilibrium, respectively, and *k*_1_ (1/min), *k*_2_ (g/(mg·min)), and *k_i_* (mg/(g·min^0.5^)) are the pseudo-first-order, pseudo-second-order, and intraparticle diffusion rate constants, respectively.

Table 9 lists the parameters obtained from the pseudo-first-order and pseudo-second-order kinetic models. The pseudo-second-order model provided a better description of all data according to the higher value of *R*^2^ and the more insignificant deviation between the fitted curve and experiment data. This suggested that chemisorption dominated the adsorption. In addition to the apparent kinetic models, we employed the intraparticle kinetic model to expound the diffusion mechanism. All curves reporting *q_t_* versus *t*^0.5^ were not a straight line passing the origin, as shown in Figure 14b. This illustrated that the intraparticle diffusion resistance was not the sole rate-limiting factor [61]. However, all curves comprised two or three consecutive linear sections, further validating that multiple continuous diffusion processes co-controlled the overall adsorption [62].

For the reusability estimation, we used the best-performing adsorbent AM41ES-60-48. The Pb(II) uptakes for the first, second, and third uses were 93.6, 87.1, and 82.3 mg/g, respectively. The decrease rate of adsorption capacity was 12% after three adsorption–desorption cycles. The uptake loss was due to some irreversible complexations between ligands and Pb(II) [63]. As shown in Table 10, the reusability of thiol-functionalized adsorbents in different studies differed dramatically, related to the difference in elution times, eluents, and the structural stability of adsorbents. The adsorbents in this study presented comparatively good reusability.

We compared the Pb(II) removal capacity of AM41ES-60-48 with other adsorbents, as shown in Table 11. Thiol-functionalized adsorbents showed a better removal capacity toward Pb(II) than those with other affinity sites. By controlling the crystallization parameters in the synthesis of Al-MCM-41, we obtained an adsorbent with a remarkable improvement in Pb(II) adsorption capacity. Although the Pb(II) removal capacity of AM41ES-60-48 was still lower than some available adsorbents due to the difference in affinity sites and carriers, this study introduced a new method of increasing Pb(II) removal capacity.

## 3. Experiments

### 3.1. Materials

Chemicals picked for this study included sodium hydroxide (NaOH), concentrated hydrochloric acid (HCl), methanol (CH_3_OH), ethanol (CH_3_CH_2_OH), glacial acetic acid (CH_3_COOH), toluene (C_7_H_8_), lead nitrate (Pb(NO_3_)_2_), ammonium chloride (NH_4_Cl), cetyltrimethylammonium bromide (CTAB), 3-mercaptopropyltrimethoxysilane (MPTMS), and 3-aminopropyltrimethoxysilane (APTMS). We purchased CTAB, MPTMS, and APTMS from Aladdin and others from Sinopharm. All chemicals were used as received without further treatment.

For a universal CFA activation strategy, we employed four kinds of CFAs. They were from four different thermal power plants of Liaoning, China, denoted as CFA1–4. CFA1–3 were ashes of the pulverized coal (PC) boiler, and CFA4 was the ash of the circulating fluidized bed (CFB) boiler.

### 3.2. CFA Activation

We used alkaline hydrothermal and alkaline fusion treatments to activate CFA in this study. Regarding alkaline hydrothermal activation, Shoppert et al. [39] optimized the atmospheric activation parameters, namely, saturate temperature, 130 g/L of NaOH, a quarter mass ratio of liquid/CFA, and 3 h of leaching duration. It improved the leaching of silica from CFA and saved energy and agents. However, they did not consider the effect of pressurized hydrothermal conditions and microwave introduction on the activation result. These methods may be helpful in the leaching of silica and alumina. Therefore, we combined the above factors to estimate the hydrothermal activation of CFA more systematically. We also used this optimal atmospheric hydrothermal condition and supplemented pressurized hydrothermal treatment and microwave-assisted treatment for comparison. More specifically, a 13 g NaOH pellet was added to 100 mL of deionized water, followed by stirring. After the NaOH pellet was thoroughly dissolved, 25 g of CFA was added. The resulting suspension was hydrothermally treated at 100 and 200 °C for 3 h while maintaining stirring at 700 rpm. The hydrothermal process proceeded in a high-pressure mini-autoclave with a precise temperature control function. After that, we decanted the suspension into a 500 mL beaker, followed by 4 h of stirring. Lastly, 5 min of centrifugation at 9000 rpm yielded the desired supernatant.

The microwave-assisted hydrothermal activation was carried out using an atmospheric microwave hydrothermal synthesizer. The activation temperature was 100 °C, and the time was 1 and 3 h. The reagent ratio was the same as the above hydrothermal process.

The alkaline fusion activation was based on the study of Shigemoto et al. [33]. They determined the optimal fusion condition for Na-X zeolite formation from a wide range of parameters. However, the fusion temperature had room for improvement regarding MCM-41 synthesis. Furthermore, no studies considered the energy consumption during the fusion process. Therefore, we changed the fusion temperature while considering energy consumption to determine a cost-effective and efficient CFA fusion temperature. More specifically, the 15 g NaOH pellet was milled to a fine powder and mixed with 12.5 g CFA. The mixture was heated at 550, 650, and 750 °C for 1 h in a high-temperature furnace. After that, we ground the sintered bulk, added the powder to 100 mL of deionized water, and agitated it for 4 h. Lastly, we obtained the desired supernatant with 9000 rpm centrifugation for 5 min.

We used the conversion ratio to determine the percentage of silica and alumina in CFAs transferred to supernatants, as defined below.
(10)$$T_{i}=\frac{100C_{i}V}{mx_{i}},$$
where *T_i_* (%) is the conversion ratio of silica or alumina, *C_i_* (mg/L) is the concentration of silica or alumina in the supernatant, *m* (g) is the mass of CFA, *V* (100 mL) is the volume of supernatant, and *x_i_* (wt.%) is the mass percentage of silica or alumina in CFA.

### 3.3. Al-MCM-41 Synthesis

Since the CFA-derived supernatant contains both silicon and aluminum ions, the skeleton of the derived MCM-41 unavoidably contains aluminum, i.e., Al-MCM-41. Compared with silica-pure MCM-41, Al-MCM-41 is more suitable for heavy-metal removal due to its high hydrothermal stability [66] and additional cation exchange capacity [67].

The procedure adopted in this study was the conventional one-pot synthesis, as described by Li et al. [68]. Briefly, 1.325 g of CTAB was added to 25 mL of deionized water followed by stirring. The resulting suspension was heated to ca. 40 °C and kept until the solution turned transparent. We mixed it with 55 mL of CFA1-derived supernatant to form the Al-MCM-41 stock solution with a Si/Al/CTAB/H_2_O molar ratio of 1:0.08:0.2:230. The solution pH was adjusted to 10.5 by adding glacial acetic acid dropwise. When the pH approached 11, the solution turned opaque, and gelation occurred. After 1 h of stirring, the gel was transferred to a Teflon-lined stainless-steel autoclave, placed in an air oven, and crystallized at 30, 60, 90, and 120 °C for 48 h. After that, we recovered the white precipitate by vacuum filtration using a copious amount of deionized water and dried it at 105 °C overnight using an air oven, obtaining the as-synthesized Al-MCM-41. In addition, we prepared Al-MCM-41 by varying aging time (24, 48, and 72 h) while keeping the hydrothermal temperature constant at 120 °C.

The template removal process followed that of Deekamwong et al. [69], except using sonication. Briefly, we added the desiccated Al-MCM-41 to 100 mL of a methanolic solution containing 2 g of NH_4_Cl in a 500 mL conical flask. Continuous stirring followed. The mixture was heated to 60 °C and then refluxed for 30 min. After that, we recovered the solid by vacuum filtration using a copious amount of ethanol. The above procedure of reflux and filtration was repeated twice to result in the surfactant-free Al-MCM-41. In addition to solvent extraction, we used calcination (550 °C, 5 h) to remove the template Al-MCM-41 crystallized at 120 °C for 48 h for comparison.

### 3.4. Functionalization

In light of the hard and soft acids and bases (HSAB) theory, we selected the thiol group for Pb(II) complexation [55]. We realized thiol grafting using a post-synthesis process, as presented by Idris et al. [9]. Briefly, the removed template of Al-MCM-41 was degassed at 150 °C for 1 h under vacuum in a vacuum oven. Next, we weighed 2 g of Al-MCM-41 and immersed it into 56 mL of a toluene solution containing 6 mL of MPTMS. The mixture was heated to 105 °C and refluxed for 4 h. After that, we recovered the solid by vacuum filtration using a copious amount of ethanol and dried it at 105 °C for overnight in an air oven, producing thiol-functionalized Al-MCM-41. For comparison, the Al-MCM-41 crystallized at 120 °C for 48 h was also grafted with APTMS to give rise to amine-functionalized Al-MCM-41. Table 12 shows the names of the Al-MCM-41 samples obtained at different crystallization conditions.

### 3.5. Characterization

X-ray fluorescence (XRF) analysis was conducted to determine the chemical composition of as-received CFAs and the activated residues using a Shimadzu ZSXPrimus II version spectrometer.

The concentrations of silica and alumina in the CFA-derived supernatants and of Pb(II) solutions were detected by the ICP technique using a PE Avio 500 type inductively coupled plasma atomic emission spectroscope (ICP-AES).

The X-ray diffraction (XRD) pattern was recorded on a Bruker D8 Advance XRD diffractometer using CuKα radiation operating at 40 kV and 200 mA, with a scanning angle 2*θ* of 10–90°, an interval of 0.02°, and a rate of 10°/min, to yield the phase structure of as-received CFAs and the activated residues.

A small-angle X-ray diffraction (SXRD) pattern was written on a PANalytical Empyrean version XRD diffractometer using CuKα radiation operating at 40 kV and 200 mA, with a scanning angle 2*θ* of 0.5–10°, an interval of 0.02°, and a rate of 1°/min, to obtain the phase structure.

Nitrogen (N_2_) adsorption–desorption isotherms were achieved using a Micromeritics ASAP 2050 adsorber. Prior to each measurement, the sample was degassed at 300 °C for 2 h. The specific surface area (*S_BET_*) was calculated using the BET method, the pore size distribution (PSD) was obtained by the BJH model, and the total pore volume (*V_PORE_*) was determined via the N_2_ amount adsorbed at a relative pressure of *p*/*p*_0_ = 0.99.

Fourier-transform infrared spectroscopy (FTIR) analysis was performed in the wavenumber range of 4000–400 cm^−1^ using an Agilent Cary 600 Series FTIR Spectrometer.

The morphology of the CFA particles was observed by scanning electron microscopy using a U8010 SEM.

The morphology of Al-MCM-41 was observed by transmission electron microscopy (TEM) using a JEM-2100 TEM.

A solid ^29^Si silica magic-angle spinning nuclear magnetic resonance (^29^Si MAS NMR) spectrum was collected on an Agilent 600 M NMR spectrometer to determine the silica’s structural coordination.

### 3.6. Pb(II) Removal

Batch experiments were carried out to determine the Pb(II) removal capacity of the adsorbents used in this study and to unveil the influence of crystallization parameters on the subsequent Pb(II) removal capacity. A 300 mg/L Pb(II) stock solution was prepared by dissolving Pb(NO_3_)_2_ in deionized water. We then obtained the desired solution by diluting the stock solution. We weighed 50 mg of adsorbent, immersed it into 50 mL of solution, and agitated it for 200 min. After that, we separated the adsorbent from the solution by centrifugation and detected the residual Pb(II) concentration of the solution by ICP. The amount of Pb(II) adsorbed was calculated as follows.
(11)$$q_{e}=\frac{(C_{0}-C_{e})V}{m},$$
where *q_e_* (mg/g) is the equilibrium adsorption capacity, *C*_0_ (mg/L) and *C_e_* (mg/L) are the initial and equilibrium Pb(II) concentration, respectively, *V* (50 mL) is the solution volume, and *m* (50 mg) is the adsorbent mass.

### 3.7. Adsorbent Regeneration

We used 0.1 mol/L HCl to elute the Pb(II) ions from the adsorbent, as described below. First, we immersed 50 mg of adsorbent into 50 mL of solution with the Pb(II) concentration of 100 mg/L. After 2 h of stirring, we recovered the adsorbent by filtration and measured the Pb(II) concentration of filtrate by ICP. The Pb(II)-loaded adsorbent was then added to 100 mL of 0.1 mol/L HCl solution to regenerate for 8 h. After that, we recovered the regenerated adsorbent and washed it using a copious amount of deionized water. The above adsorption–desorption process was repeated three times to estimate the reusability of the adsorbent.

## 4. Conclusions

Four CFAs from different coals with various formation conditions were activated by alkaline hydrothermal and alkaline fusion treatments to obtain amorphous Si–Al leaches. By comparing two activation strategies and selecting different activation temperatures, we can draw three conclusions. First, conventional hydrothermal activation can extract the amorphous aluminosilicate glasses, but the leaching ability is not high (33–67% for silica and 13–23% for alumina). Second, microwave-assisted hydrothermal activation not only significantly speeds up the leaching of silica and alumina but also improves the mass of leaches. The silica mass leached from CFA2 is more than the mass of amorphous silicate glasses in CFA2, indicating that it derives silica from crystals. Third, alkaline fusion activation can derive silica and alumina from both the amorphous aluminosilicates and the crystals. Therefore, the conversion rate is significantly higher than that of conventional hydrothermal activation. Furthermore, we determined the optimal fusion temperature, 650 °C, which yields the maximal mass of Al-MCM-41 when consuming the same energy and has universality for all CFAs.

The Al-MCM-41 synthesized at different crystallization temperatures and times varied significantly in pore structure and surface hydroxyl density. As crystallization temperature increased at 100 °C intervals or aging time increased at 24 h intervals, we obtained an Al-MCM-41 with higher uniformity, a thicker pore wall, a lower specific area, and fewer hydroxyl groups. These differences presented a significant effect on the Pb(II) removal capacity upon thiol-functionalization. AM41ES-60-48 achieved the maximal adsorption capacity, 153.0 mg/g. This work, therefore, introduces a means of fortifying the uptake toward Pb(II) and suggests a low-temperature synthesis of adsorbents, which consumes less energy.

Either the Langmuir or the Freundlich model can adequately describe the experimental data, depending on the distribution uniformity of affinity sites. The thermodynamic study suggested a spontaneous and endothermic adsorption process. Upon adsorption, the randomness of Pb(II) increased, despite losing a degree of freedom. The pseudo-second-order kinetic model fit all adsorption processes better, indicative of chemically dominated adsorption. The intraparticle diffusion model indicated that several consecutive diffusion steps co-controlled the overall adsorption rate.

## Figures and Tables

**Figure 1 ijms-22-06540-f001:**
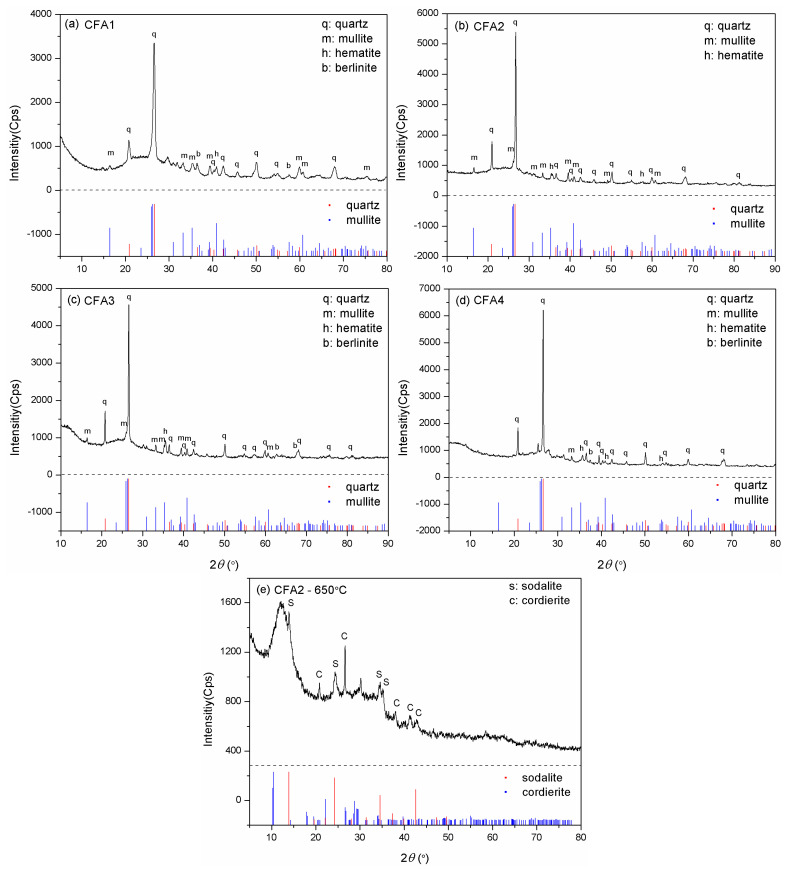
XRD patterns for (**a**) CFA1, (**b**) CFA2, (**c**) CFA3, (**d**) CFA4, and (**e**) CFA2 residue after fusion at 650 °C.

**Figure 2 ijms-22-06540-f002:**
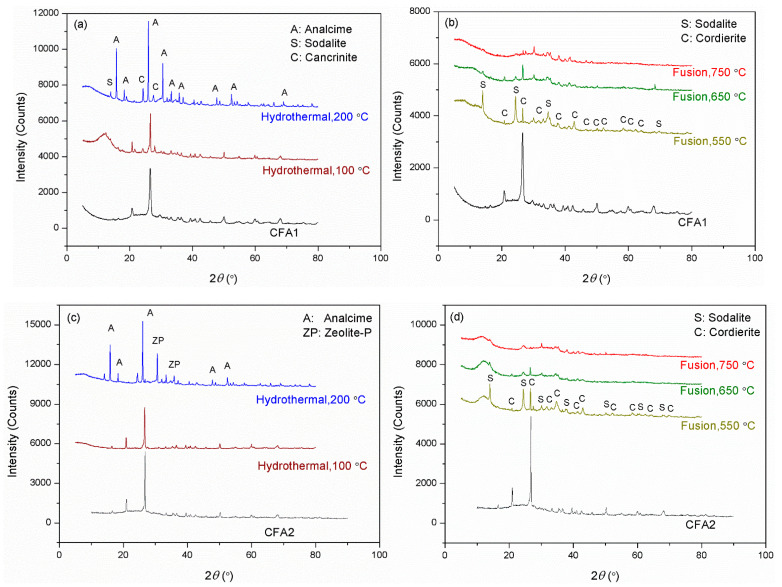
XRD patterns for the residues after hydrothermal treatments of (**a**) CFA1, (**c**) CFA2, (**e**) CFA3, and (**g**) CFA4, after microwave-assisted hydrothermal treatments of (**i**) CFA1, and after fusion treatments of (**b**) CFA1, (**d**) CFA2, (**f**) CFA3, and (**h**) CFA4.

**Figure 3 ijms-22-06540-f003:**
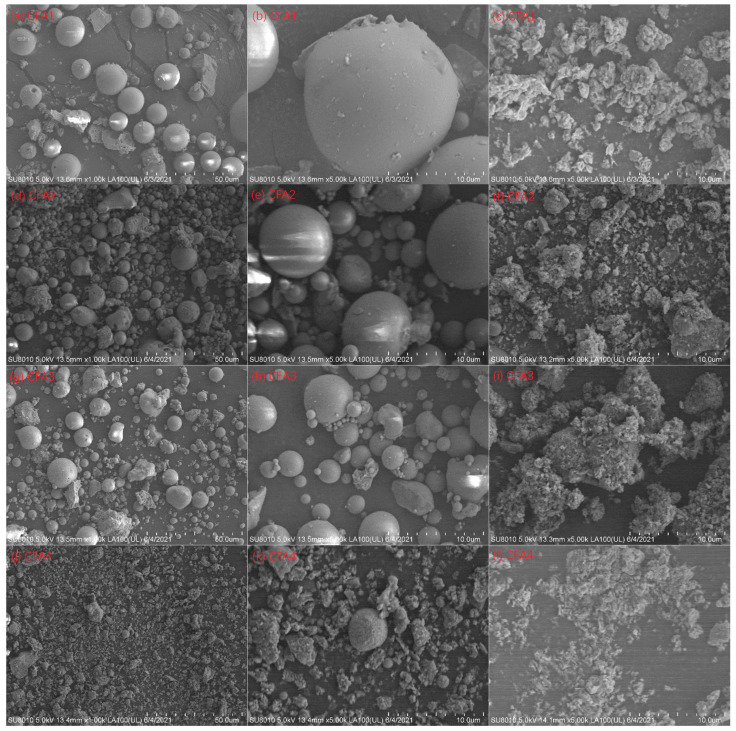
SEM images at 1000× magnification: (**a**) CFA1, (**d**) CFA2, (**g**) CFA3, and (**j**) CFA4; 5000× magnification: (**b**) CFA1, (**e**) CFA2, (**h**) CFA3, and (**k**) CFA4; 5000× magnification: residues of (**c**) CFA1, (**f**) CFA2, (**i**) CFA3, and (**l**) CFA4 after 550 °C of fusion.

**Figure 4 ijms-22-06540-f004:**
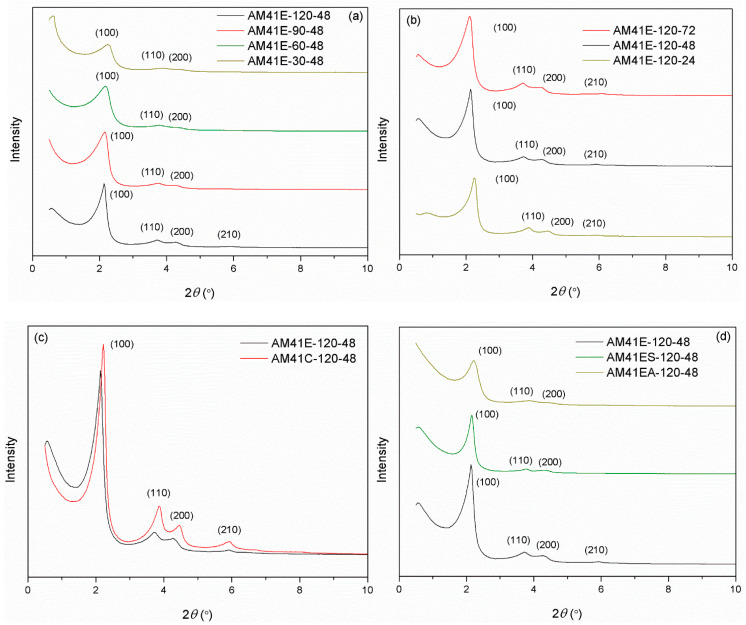
SXRD patterns for (**a**) AM41E-120-48, AM41E-90-48, AM41E-60-48, and AM41E-30-48, (**b**) AM41E-120-72, AM41E-120-48, and AM41E-120-24, (**c**) AM41E-120-48 and AM41C-120-48, and (**d**) AM41E-120-48, AM41ES-120-48, and AM41EA-120-48.

**Figure 5 ijms-22-06540-f005:**
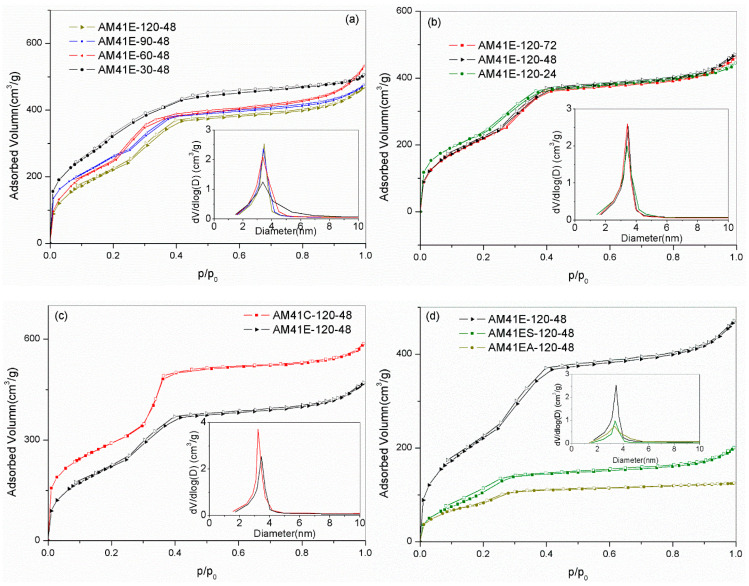
N_2_ adsorption–desorption isotherms for (**a**) AM41E-120-48, AM41E-90-48, AM41E-60-48, and AM41E-30-48 (+50), (**b**) AM41E-120-72, AM41E-120-48, and AM41E-120-24, (**c**) AM41E-120-48 and AM41C-120-48, and (**d**) AM41E-120-48, AM41ES-120-48, and AM41EA-120-48.

**Figure 6 ijms-22-06540-f006:**
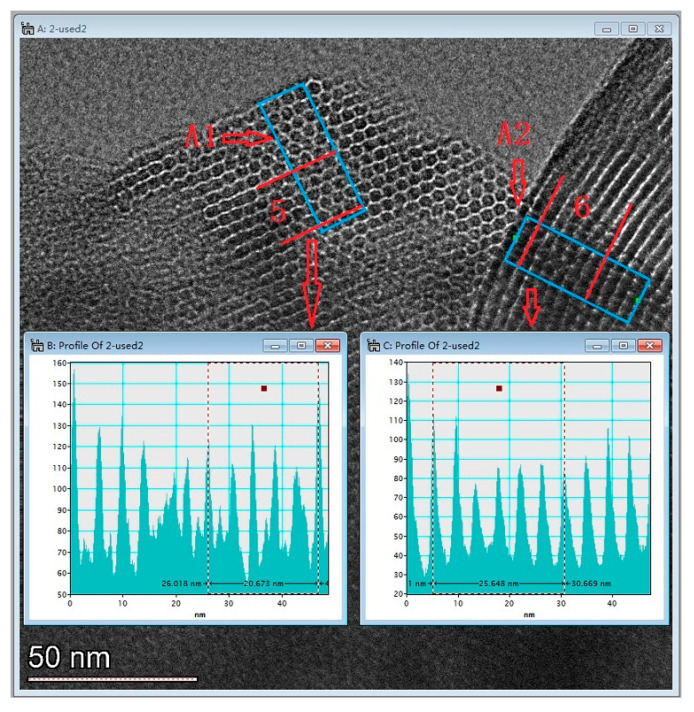
TEM images of AM41E-120-48.

**Figure 7 ijms-22-06540-f007:**
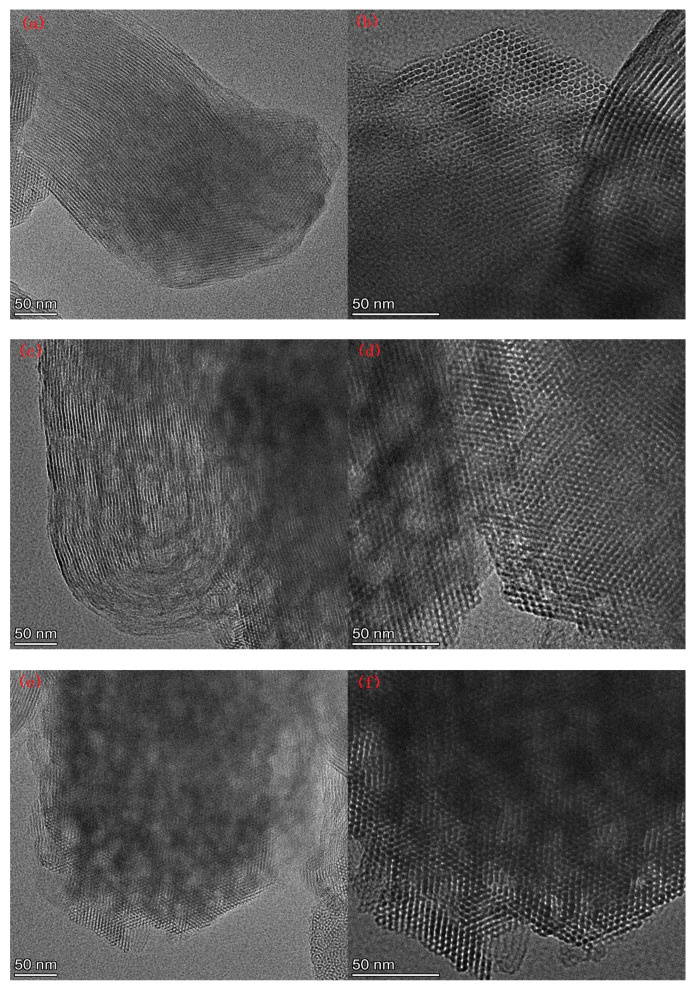
SEM images for (**a**,**b**) AM41E-120-48, (**c**,**d**) AM41E-90-48, (**e**,**f**) AM41E-60-48, (**g**,**h**) AM41E-30-48, (**i**,**j**) AM41E-120-72, and (**k**,**l**) AM41E-120-24.

**Figure 8 ijms-22-06540-f008:**
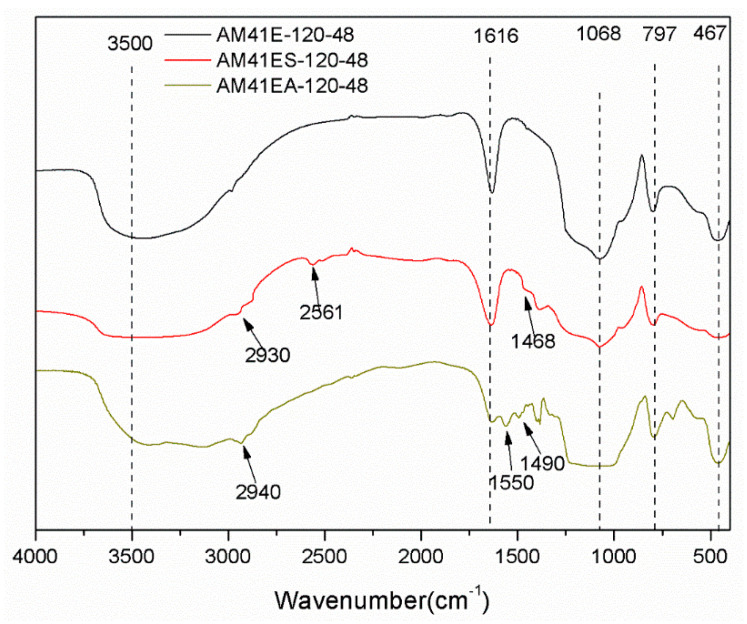
FTIR patterns for AM41E-120-48, AM41ES-120-48, and AM41EA-120-48.

**Figure 9 ijms-22-06540-f009:**
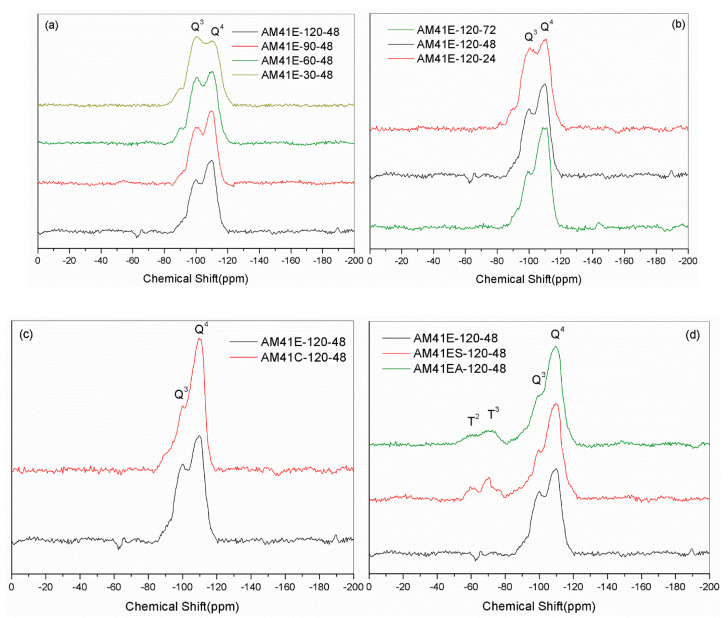
^29^Si MAS NMR spectra for (**a**) AM41E-120-48, AM41E-90-48, AM41E-60-48, and AM41E-30-48, (**b**) AM41E-120-72, AM41E-120-48, and AM41E-120-24, (**c**) AM41E-120-48 and AM41C-120-48, and (**d**) AM41E-120-48, AM41ES-120-48, and AM41EA-120-48.

**Figure 10 ijms-22-06540-f010:**
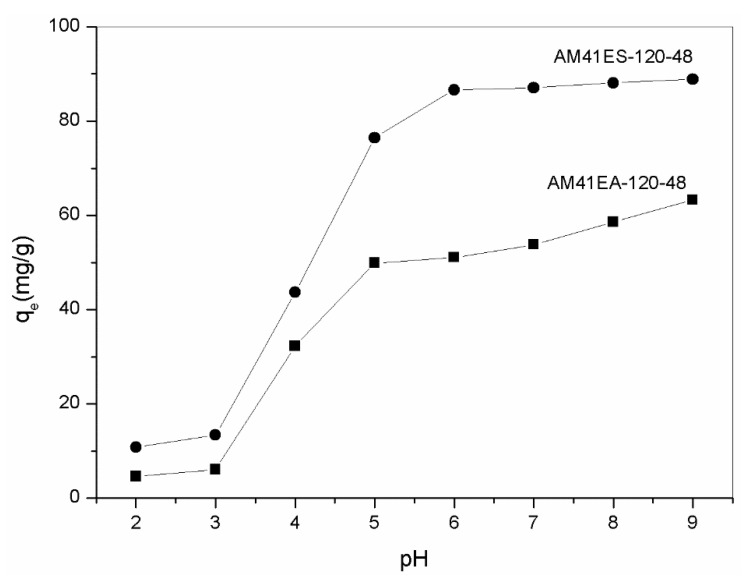
Adsorption capacity of AM41ES-120-48 and AM41EA-120-48 for Pb(II) at different pH. Experimental conditions: adsorption temperature *T* = 25 °C, adsorbent dose *m* = 50 mg, solution volume *V* = 50 mL, contact time *t* = 200 min, pH = 2, 3, 4, 5, 6, 7, 8, and 9, and initial Pb(II) concentration *C*_0_ = 100 mg/L.

**Figure 11 ijms-22-06540-f011:**
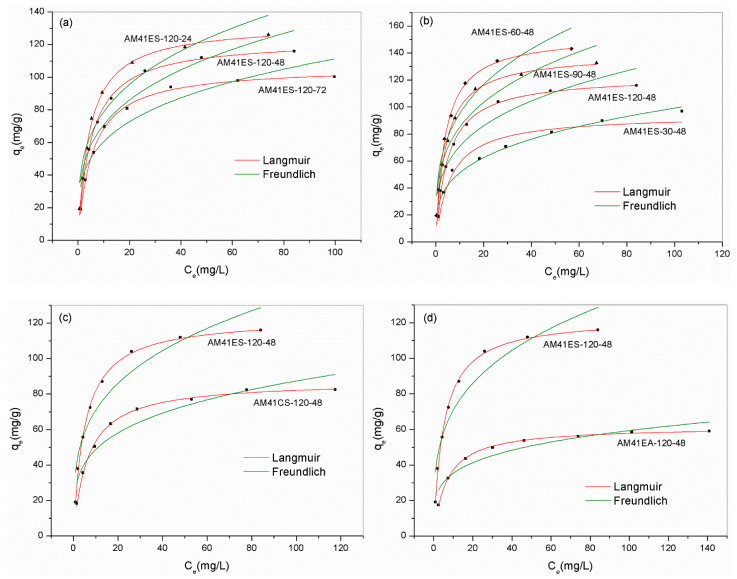
Experiment data for Pb(II) removal and fitted isotherms for (**a**) AM41ES-120-72, AM41ES-120-48, and AM41ES-120-24, (**b**) AM41ES-120-48, AM41ES-90-48, AM41ES-60-48, and AM41ES-30-48, (**c**) AM41ES-120-48 and AM41CS-120-48, and (**d**) AM41ES-120-48 and AM41EA-120-48. Experimental condition**s**: adsorption temperature *T* = 25 °C, adsorbent dose *m* = 50 mg, solution volume *V* = 50 mL, contact time *t* = 200 min, pH = 6, and initial Pb(II) concentration *C*_0_ = 20, 40, 60, 80, 100, 130, 160, and 200 mg/L.

**Figure 12 ijms-22-06540-f012:**
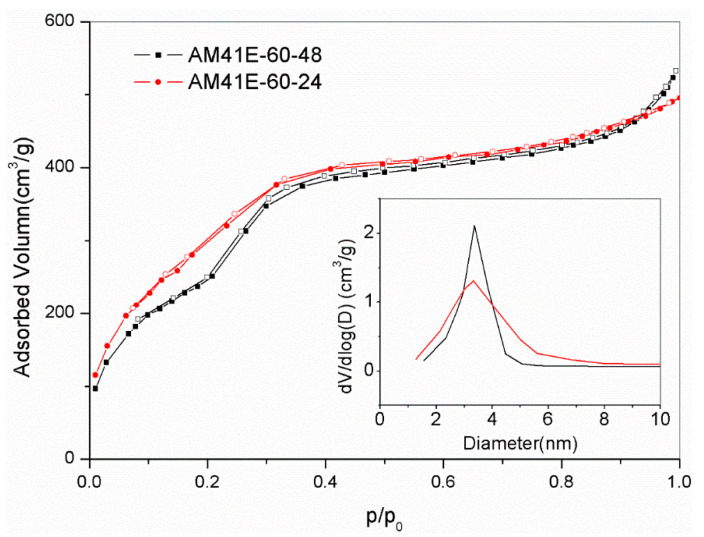
N_2_ adsorption–desorption isotherms for AM41E60-48 and AM41E60-24.

**Figure 13 ijms-22-06540-f013:**
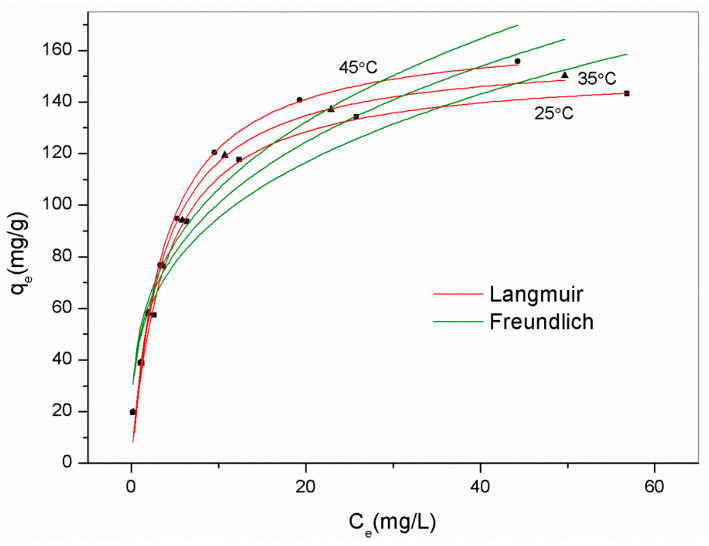
Experimental data for Pb(II) removal by AM41ES-60-48 at different temperatures and fitted isotherms. Experimental condition**s**: adsorption temperature *T* = 25, 35, and 45 °C, adsorbent dose *m* = 50 mg, solution volume *V* = 50 mL, contact time *t* = 200 min, pH = 6, and initial Pb(II) concentration *C*_0_ = 20, 40, 60, 80, 100, 130, 160, and 200 mg/L.

**Figure 14 ijms-22-06540-f014:**
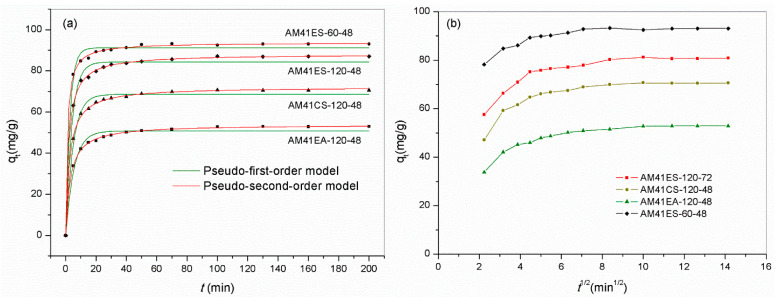
Adsorption curves reporting Pb(II) adsorption capacity versus time by AM41ES-120-48, AM41CS-120-48, AM41EA-120-48, and AM41ES-60-48, and fitted curves by (**a**) pseudo-first-order kinetics and pseudo-second-order kinetics and (**b**) intraparticle diffusion kinetics. Experimental condition**s**: adsorption temperature *T* = 25 °C, adsorbent dose *m* = 50 mg, solution volume *V* = 50 mL, contact time *t* = 5, 10, 15, 20, 25, 30, 40, 50, 70, 100, 130, 160, and 200 min, pH = 6, and initial Pb(II) concentration *C*_0_ = 100 mg/L.

**Table 1 ijms-22-06540-t001:** Chemical composition (wt.%) and mineral composition (wt.%) of CFAs used in this study.

Samples	Chemical Composition						Mineral Composition			
	SiO_2_	Al_2_O_3_	Fe_2_O_3_	CaO	K_2_O	MgO	Crystal Phase		Glass Phase	
							Quartz	Mullite	SiO_2_	Al_2_O_3_
CFA1	62.7	19.8	3.8	6.4	2.2	1.0	38.3	19.6	18.9	5.7
CFA2	63.2	20.9	4.4	5.7	2.5	1.1	44.7	21.7	12.4	5.3
CFA3	64.3	22.7	5.0	2.6	2.5	1.1	43.5	25.3	13.7	4.5
CFA4	58.9	21.3	9.1	5.3	2.4	1.1	41.1	9.1	16.2	12.1

**Table 2 ijms-22-06540-t002:** CFA activation methodologies, treatment parameters, and results.

CFA	Activation Methods and Parameters					Supernatant		Conversion (%)	
	Method	Temp (°C)	Time (h)	Na/ CFA	L/S	Si (mg/L)	Al (mg/L)	Si (%)	Al (%)
CFA1	Hydro	100	3	0.52	4	9303	347	12.7	1.32
		200				6153	260	8.4	0.99
	Microwave Hydro	100	1			11,281	632	15.4	3.21
		100	3			15,603	825	21.3	4.19
	Fusion	550	1	1.25	8	7314	428	20.0	3.26
		650				9266	702	25.3	5.35
		750				9706	806	26.5	6.14
CFA2	Hydro	100	3	0.52	4	4351	341	5.9	1.24
		200				8260	118	11.2	0.43
	Fusion	550	1	1.25	8	7426	373	20.1	2.71
		650				9145	579	24.8	4.21
		750				9919	968	26.9	7.04
CFA3	Hydro	100	3	0.52	4	5700	291	7.6	0.97
		200				10,650	36	14.2	0.12
	Fusion	550	1	1.25	8	8038	380	21.4	2.53
		650				9375	575	25	3.83
		750				10,238	909	27.3	6.06
CFA4	Hydro	100	3	0.52	4	3644	449	5.3	1.59
		200				6600	93	9.6	0.33
	Fusion	550	1	1.25	8	7697	199	22.4	1.41
		650				9453	377	27.5	2.67
		750				9900	542	28.8	3.84

**Table 3 ijms-22-06540-t003:** Analysis of energy consumption per mass of Al-MCM-41 produced following alkaline fusion of CFA1 and 4.

Energy Consumption of the Alkaline Fusion of CFA1 and 4						
**(1)** **Computation of the Mass of Al-MCM-41 Produced**						
(a) Ash	CFA1			CFA4		
(b) Ash type	PC ash			CFB ash		
(c) CFA mass: *m_CFA_* (kg)	1000			1000		
(d) CFA specific heat capacity at constant pressure: *C_p_* (kJ/(kg·K))	0.92			0.92		
(e) SiO_2_ wt.% in CFA: *x_Si_* (%)	62.7			58.9		
(f) Al_2_O_3_ wt.% in CFA: *x_Al_* (%)	19.8			21.3		
(g) Fusion temperature: *T_fusion_* (°C)	Different *T_fusion_*					
	550	650	750	550	650	750
(h) SiO_2_ conversion: *E_Si_* (%)	*E_Si_* obtained at different *T_fusion_*					
	20	25.3	26.5	22.4	27.5	28.8
(i) Al_2_O_3_ conversion: *E_Al_* (%)	*E_Al_* obtained at different *T_fusion_*					
	3.26	5.35	6.14	1.41	2.67	3.84
(j) Provided that all silica and alumina substances transferred from CFA to supernatants crystallize into Al-MCM-41, the mass of synthesized Al-MCM-41 has the following approximate expression: $m_{M41}\approx m_{CFA}(\frac{x_{Si}}{100}\frac{E_{Si}}{100}+\frac{x_{Al}}{100}\frac{E_{Al}}{100})$						
(k) Al-MCM-41 mass: *m_M41_* (kg)	*m*_*M*41_ obtained at different *T_fusion_*					
	132	169	178	135	168	178
**(2)** **Analysis of Energy Consumption**						
(a) Effective heat *Q_e_* is the energy heating CFA from room temperature to the specified fusion temperature. If the room temperature is 20 °C, it has the expression $Q_{e}=m_{CFA}C_{p}(T_{fusion}-20)$.						
Effective heat: *Q_e_* (kJ)	487,600	579,600	671,600	487,600	579,600	671,600
(b) Energy loss *Q*_1_ is only related to the number of furnace doors. The power loss per door is 2 kW, and the furnace has two doors; thus, the energy loss *Q*_1_ for 1 h of activation is 14,400 kJ.						
Energy loss *Q*_1_ (kJ)	14,400	14,400	14,400	14,400	14,400	14,400
(c) Energy loss *Q*_2_ is only related to the number of furnace top fans. The power loss per fan is 1 KW, and the furnace has two top fans; thus, the energy loss *Q*_2_ for 1 h of activation is 7200 kJ.						
Energy loss *Q*_2_ (kJ)	7200	7200	7200	7200	7200	7200
(d) Energy loss *Q*_3_ is only related to the number of electric heating tubes. The power loss per heating tube is 1 KW, and the furnace has nine tubes; thus, the energy loss *Q*_3_ for 1 h of activation is 32,400 KJ.						
Energy loss *Q*_3_ (kJ)	32,400	32,400	32,400	32,400	32,400	32,400
(e) Energy loss *Q*_4_ is only related to the number of stacking devices. The power loss per stacking device is 1.5 KW, and the furnace has four stacking devices. Provided that the working time of stacking devices is 10 min, the energy loss *Q*_4_ is 3600 kJ.						
Energy loss *Q*_4_ (kJ)	3600	3600	3600	3600	3600	3600
(f) Energy loss *Q*_5_ is the energy loss caused by the observation window, thermocouple, and gas in the furnace. The power loss is generally 2 KW, and the energy loss *Q*_5_ for 1 h of activation is 7200 kJ.						
Energy loss *Q*_5_ (kJ)	7200	7200	7200	7200	7200	7200
(g) Energy loss *Q*_6_ is the heat loss through the wall of the furnace. When the furnace size is fixed (assuming the total outside wall surface area *A* is 30 m^2^), energy loss *Q*_6_ is only related to the outside wall surface temperature. Generally, it increases slightly as the furnace temperature rises. When the furnace temperature is 550, 650, and 750 °C, the outside wall surface temperature is 55, 60, and 65 °C, respectively, and the heat dissipation coefficient k is then 0.4, 0.45, and 0.5 kW/m^2^, respectively. Accordingly, we can determine the energy loss *Q*_6_ via the expression $Q_{6}=3600kA$.						
Energy loss *Q_6_* (kJ)	43,200	48,600	54,000	43,200	48,600	54,000
(h) Total energy consumption *Q_all_* is equal to the sum of effective heat ^Q_e_^ and various energy losses multiplied by the coefficient α (=1.2), and it has the expression $Q_{all}=\alpha (Q_{e}+\sum_{i=1}^{6}Q_{i})$ .						
Total energy consumption *Q_all_* (kJ)	714,720	831,600	948,480	714,720	831,600	948,480
(i) Total energy loss *Q_loss_* is the difference between total energy consumption *Q_all_* and effective heat *Q_e_*.						
Total energy loss *Q_loss_* (kJ)	227,120	252,000	276,880	227,120	252,000	276,880
**(3)** **Results**						
(a) Total efficiency *f* is the ratio of effective heat *Q_e_* to total energy consumption *Q_all_.*						
Total efficiency *f* (%)	68.2	69.7	70.8	68.2	69.7	70.8
(b) Energy consumption rate *q* is the energy consumption per mass of Al-MCM-41 produced. It has the expression $q=\frac{Q_{all}}{m_{M41}}$.						
Energy consumption rate *q* (kJ/kg)	5421	4914	5319	5297	4960	5334

**Table 4 ijms-22-06540-t004:** Extraction effect of silica and alumina from CFA using alkaline fusion activation in various references.

Reference	CFA		Activation Methods and Parameters				Supernatant		Conversion (%)	
	Si (wt.%)	Al (wt.%)	Temperature (°C)	Time (h)	Alkali/ CFA	Water/ CFA	Si (mg/L)	Al (mg/L)	Si (%)	Al (%)
Izabela et al. [35]	46–55	18–32	550	1	1.2	4	1400–2900	960–2300	7–10	5–12
Halina et al. [43]	66	16				5	10,000	367	33	4
Chang et al. [44]	56	31				5	2740	528	8	3
Kumar et al. [31]	67	19				4	11,000	380	18	2

**Table 5 ijms-22-06540-t005:** Pore structural properties of the samples investigated in this study.

Sample	SXRD		N_2_ Isotherms			
	*d*_100_, nm	*a*_0_, nm	*S_BET_*, m^2^/g	*d_BJH_*, nm	*V_PORE_*, cm^3^/g	W, nm
AM41E-120-48	4.14	4.78	791	3.47	0.77	1.31
AM41E-90-48	4.10	4.73	803	3.42	0.78	1.29
AM41E-60-24	-	-	848	3.46	0.81	1.22
AM41E-60-48	4.06	4.69	824	3.44	0.87	1.25
AM41E-30-48	3.95	4.56	861	3.39	0.75	1.17
AM41C-120-48	3.98	4.60	1055	3.24	0.96	1.36
AM41E-120-24	4.02	4.64	831	3.38	0.73	1.26
AM41E-120-72	4.17	4.82	788	3.49	0.76	1.33
AM41ES-120-48	4.10	4.73	374	3.38	0.33	-
AM41EA-120-48	3.98	4.60	261	3.21	0.21	-

**Table 6 ijms-22-06540-t006:** Deconvolution analysis of ^29^Si NMR results.

Sample	Q^4^, ppm	Q^3^, ppm	Area of Q^4^	Area of Q^3^	Q^4^/Q^3^
AM41E-120-48	−112	−102	21,143	11,746	1.8
AM41E-90-48	−111	−102	20,819	14,871	1.4
AM41E-60-48	−111	−102	18,751	15,626	1.2
AM41E-30-48	−112	−102	17,283	19,203	0.9
AM41C-120-48	−112	−101	23,102	5635	4.1
AM41E-120-24	−112	−102	19,885	15,296	1.3
AM41E-120-72	−112	−101	21,963	6863	3.2
AM41ES-120-48	−111	−102	18,726	4161	4.5
AM41EA-120-48	−112	−102	18,931	4403	4.3

**Table 7 ijms-22-06540-t007:** Element composition (wt.%) of adsorbents investigated in this study.

Adsorbent	S	C	H	Si1 ^b^	Si2 ^b^	Al	O	Si2/Al ^c^	S/Pb ^c^
AM41ES-120-72	6.95	8.00	2.50	6.38	30.3	2.26	38.2	12.9	4.22
AM41ES-120-48	8.10	9.16	2.77	6.97	28.4	2.15	35.8	12.8	4.26
AM41ES-120-24	8.80	9.59	3.04	7.38	29.0	2.13	35.5	13.1	4.30
AM41ES-90-48	9.32	10.99	3.42	8.31	27.4	2.10	33.7	12.8	4.32
AM41ES-60-48	10.32	11.28	3.62	9.32	26.2	1.99	33.0	12.9	4.36
AM41ES-30-48	7.03	8.16	2.56	5.86	30.1	2.17	37.8	13.4	4.42
AM41CS-120-48	5.60	6.55	1.98	5.01	31.9	2.33	40.0	13.1	4.15
AM41EA-120-48 ^a^	3.95	9.68	2.85	7.85	30.8	2.30	38.7	12.9	9.49

^a^ The numbers 3.95 and 9.49 are the weight percentage of N and molar ratio of N/Pb. ^b^ Si1 is the silica in organosilanes, and Si2 is the silica in the skeleton of Al-MCM-41. ^c^ Si2/Al and S/Pb are both molar ratios.

**Table 8 ijms-22-06540-t008:** Characteristic parameters and constants of Langmuir and Freundlich models and the correlation coefficients *R*^2^.

Adsorbent (Temperature)	Langmuir			Freundlich		
	*q_m_*, mg/g	*b*, L/mg	*R* ^2^	*k_F_*, mg/g	*n*	*R* ^2^
AM41ES-120-72 (25)	106.4	0.181	0.997	32.9	3.783	0.890
AM41ES-120-48 (25)	123.0	0.198	0.998	37.1	3.567	0.896
AM41ES-120-24 (25)	132.4	0.226	0.996	41.2	3.563	0.896
AM41ES-90-48 (25)	139.5	0.247	0.995	44.3	3.534	0.906
AM41ES-60-48 (25)	153.0	0.262	0.992	48.3	3.397	0.907
AM41ES-60-48 (35)	159.2	0.275	0.991	49.9	3.278	0.928
AM41ES-60-48 (45)	167.3	0.269	0.988	51.5	3.175	0.935
AM41ES-30-48 (25)	94.1	0.163	0.943	26.4	3.476	0.966
AM41CS-120-48 (25)	87.3	0.157	0.997	27.1	3.938	0.892
AM41EA-120-48 (25)	61.6	0.153	0.998	21.0	4.436	0.886

**Table 9 ijms-22-06540-t009:** Kinetic constants and *R*^2^ of pseudo-first-order and pseudo-second-order models for AM41ES-120-48, AM41CS-120-48, AM41EA-120-48, and AM41ES-60-48.

Adsorbent	Pseudo-First-Order Kinetics		Pseudo-Second-Order Kinetics	
	*k*_1_, 1/min	*R* ^2^	*k*_2_, g/(mg·min)	*R* ^2^
AM41ES-120-48	0.251	0.984	0.0059	0.999
AM41CS-120-48	0.209	0.986	0.0056	0.999
AM41EA-120-48	0.186	0.976	0.0063	0.999
AM41ES-60-48	0.366	0.991	0.0103	0.999

**Table 10 ijms-22-06540-t010:** Comparison of regeneration capacity of thiol-functionalized adsorbents.

Adsorbent	Cycles	Time (h)	Eluent	Capacity Loss (%)	Reference
Thiol-functionalized MCM-41	3	-	Concentrated HCl	65	[64]
Thiol-functionalized MCM-41	7	30	1 mol/L of HNO_3_	8	[50]
Thiol-functionalized MCM-41	3	6	0.1 mol/L triethylenetetramine	32	[63]
Magnetic thiolated -chitosan composites	5	-	0.1 mol/L HCl	7	[59]
AM41ES-60-48	3	8	0.1 mol/L HCl	12	This study

**Table 11 ijms-22-06540-t011:** Comparison of AM41ES-60-48 with other adsorbents in the removal capacity of Pb(II).

Adsorbent	Capacity (mg/g)	Reference
Thiol-functionalized MCM-41	110.6	[50]
Hollow SiO_2_ microspheres with thiol-rich surfaces	125.3	[12]
Thiol-functionalized silica	117.5	[58]
Thiol-functionalized magnetic mesoporous silica	91.5	[60]
Thiol-functionalized large pore diameter MCM-41	88.0	[9]
Post-NH_2_-extracted MCM-41	64.21	[48]
Magnetically modified mesoporous nanoparticles	238.1	[65]
Chitosan-functionalized MCM-41-A	90.9	[2]
AM41ES-60-48	153.0	This study

**Table 12 ijms-22-06540-t012:** Nomenclature of samples synthesized at different crystallization parameters.

Temp (°C)	Time (h)	Detemplation	Al-MCM-41	Ligand	Adsorbent
30	48	Extraction	AM41E-30-48	MPTMS	AM41ES-30-48
60	24	Extraction	AM41E-60-24	MPTMS	AM41ES-60-24
60	48	Extraction	AM41E-60-48	MPTMS	AM41ES-60-48
90	48	Extraction	AM41E-90-48	MPTMS	AM41ES-90-48
120	24	Extraction	AM41E-120-24	MPTMS	AM41ES-120-24
120	48	Extraction	AM41E-120-48	MPTMS	AM41ES-120-48
120	48	Extraction	AM41E-120-48	APTMS	AM41EA-120-48
120	48	Calcination	AM41C-120-48	MPTMS	AM41CS-120-48
120	72	Extraction	AM41E-120-72	MPTMS	AM41ES-120-72

## Data Availability

The data presented in this study are available in article.

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
