# Peer review of "Efficient Activation of Coal Fly Ash for Silica and Alumina Leaches and the Dependence of Pb(II) Removal Capacity on the Crystallization Conditions of Al-MCM-41"

_ijms, 2021, doi:10.3390/ijms22126540_

Round 1

Reviewer 1 Report

Manuscript entitled “Efficient activation of coal fly ash for silica and alumina leaches and dependence of Pb(II) removal capacity on crystallization conditions of Al-MCM-41” raises an interesting and important topic of heavy metals’ removal from the environment. However, the presented manuscript is written in somewhat chaotic way which makes it difficult for fully understanding.

Below, please find my detailed comments:

  • English in the whole article need to be improved. Some of the presented sentence are definitely too long, thus the potential reader will get lost.
  • Introduction, paragraph 2 (line 35). This is the continuation of previous paragraph, in my opinion it should not be presented as separated one.
  • All the shortcuts need to be explained when they appear in text for the first time (line 48-49). What is the difference between the presented silica species?
  • It is commendable to use the waste material as a raw material in presented research. However, Authors not very clearly marked the novelty of their work, and a frequently repeated phrase: “the research procedure was taken from XXX et al.” makes it even harder to spot this novelty.
  • Figure 2: why the XRD peaks for samples after fusion treatment are not marked?
  • Table 4 is completely illegible.
  • Figures 10 d and 13 b are upside down for some reason.
  • Figure 9: What was the adsorption capacity of AM41ES-120-48 and AM41EA-120-48 above pH 7? Did Authors checked this?
  • Did Authors checked the reusability of adsorbents in more than one cycle?

Concluding, I suggest major revision of presented manuscript.

Author Response

Dear Professor:

Thank you so much for your valuable suggestions. I have revised them all. Please see the attachment.

Reviewer 2 Report

Efficient activation of coal fly ash for silica and alumina leaches and dependence of Pb(II) removal capacity on crystallization conditions of Al-MCM-41 is very interesting paper! Some changes are required.

Please to write full name:

Line 9, 10: Al-MCM-41 (Aluminium-Mobil Composition of Matter No. 41?), CFAs (coal flying ashes?)

Line 14: SXRD, FTIR, 29Si NMR, and TEM

Line 48: please to write full names: SBA-15, MCM-41, MCM-48, HMS, 

Line 74: During the synthesis, the crystallization process after the MCM- 74
41 stock solution is prepared is a crucial step (In which reactor was performed crystallization? Which temperature range was studied for Crystallization?

Line 81: As a result, the heavy metals (such as Pb, Ni, Co) removal

Line 269: What is the content for Yttrium and vanadium in CFA in TAble 2?

Line 298: amorphous aluminosilicates from CFAs but cannot open the structure of quartz ...( what is ratio of amorphous aluminosilicates in CFA? You did not mention it during discussion of XRD-Analysis!)

Line 309: Therefore, the extraction effect of pressurized alkaline hydrothermal activation of CFA (at 200°C? which pressure?)

 Line 395; Because high crystallization temperature (value?) and long aging time (value?)

Conclusion:

Line 634: As crystallization temperature or aging time increases (in which interval?)

General comments

Line 292:The maximal Conversion of Si ( 28.8) and Al (7%) is presented In Table 3. CFA activation methodologies, treatment parameters, and results. Why? How to improve it in order to reach maximal conversion of Al and Si (more than 95 %)

Do you have one SEM-analysis of CFA . What is morphology and particle size of CFA?

 Line 626, 627: Four CFAs from different coals and having various formation conditions are activated by alkaline hydrothermal and alkaline fusion treatments to get amorphous leaches. Maybe microwave hydrothermal treatment in short time can better destroy mullite and maximise conversion of Al and SI.

Author Response

(The authors gave the same response as above.)

Reviewer 3 Report

The manuscript presents CFA activation strategy by varying CFA types, activation methods, and process parameters. Alkaline hydrothermal and alkaline fusion are two treatments used to CFA activation. The CFA derived supernatants were crystallized into Al-MCM-41 at different temperatures and times. All the materials were characterized by different techniques. Finally, the Pb (II) removal capacity from aqueous solution was investigated.

The work is interesting and well organized. However, some minor revisions are required:

  • Please check Figure 10 d and Figure 13 b. They are set upside down.
  • I suggest you insert in the introduction paragraph this reference [Embodied energy as key parameter for sustainable materials selection: The case of reusing coal fly ash for removing anionic surfactants. Journal of Cleaner Production 141 (2017) 230-236. http://dx.doi.org/10.1016/j.jclepro.2016.09.070]. This work shows the application of coal fly ash as adsorbent for anionic surfactants.

Author Response

(The authors gave the same response as above.)

Round 2

Reviewer 1 Report

Authors answered all my questions and doubts as well as significantly improved the manuscript. In this regard I suggest acceptance of presented article.